



# Representing Low-Intensity Fire Sensible Heat Output in a Mesoscale Atmospheric Model with a Canopy Submodel: A Case Study with ARPS-CANOPY (version 5.2.12)

Michael T. Kiefer[1], Warren E. Heilman[2], Shiyuan Zhong[1], Joseph J. Charney[2], Xindi Bian[2], Nicholas S. Skowronski[3], Kenneth L. Clark[4], Michael R. Gallagher[4], John L. Hom[2], and Matthew Patterson[3]

[1]Department of Geography, Environment, and Spatial Sciences, Michigan State University, East Lansing, MI 48824, USA
[2]USDA Forest Service, Northern Research Station, Lansing, MI 48910, USA
[3]USDA Forest Service, Northern Research Station, Morgantown, WV 26505, USA
[4]USDA Forest Service, Northern Research Station, New Lisbon, NJ 08064, USA

**Correspondence:** Michael T. Kiefer (mtkiefer@msu.edu)

**Abstract.** Mesoscale models are a class of atmospheric numerical model designed to simulate atmospheric phenomena with horizontal scales of about 2-200 km, although they are also applied to microscale phenomena, with horizontal scales less than about 2 km. Mesoscale models are capable of simulating wildland fire impacts on atmospheric flows if combustion by-products (e.g., heat, smoke) are properly represented in the model. One of the primary challenges encountered in applying a mesoscale

model to studies of fire-perturbed flows is the representation of the fire sensible heat source in the model. Two primary methods have been implemented previously: turbulent sensible heat flux, either in the form of an exponentially-decaying vertical heat flux profile or surface heat flux; and soil temperature perturbation. In this study, the ARPS-CANOPY model, a version of the Advanced Regional Prediction System (ARPS) model with a canopy submodel, is utilized to simulate the turbulent atmosphere during a low-intensity operational prescribed fire in the New Jersey Pine Barrens. The study takes place in two phases: model

assessment and model sensitivity. In the model assessment phase, analysis is limited to a single control simulation in which the fire sensible heat source is represented as an exponentially-decaying vertical profile of turbulent sensible heat flux. In the model sensitivity phase, a series of simulations are conducted to explore the sensitivity of model-observation agreement to (i) the method used to represent the fire sensible heat source in the model and (ii) parameters controlling the magnitude and vertical distribution of the sensible heat source. In both phases, momentum and scalar fields are compared between the

model simulations and data obtained from six flux towers located within and adjacent to the burn unit. The multi-dimensional model assessment confirms that the model reproduces the background and fire-perturbed atmosphere as depicted by the tower observations, although the model underestimates the turbulent kinetic energy at the top of the canopy at several towers. The model sensitivity tests reveal that the best agreement with observations occurs when the fire sensible heat source is represented as a turbulent sensible heat flux profile, with surface heat flux magnitude corresponding to the peak 1-min mean observed heat flux averaged across the flux towers, and an e-folding extinction depth corresponding to the average canopy height in

the burn unit. The study findings provide useful guidance for improving the representation of the sensible heat released from low-intensity prescribed fires in mesoscale models.



## 1 Introduction

Studies of wildland fire-perturbed atmospheric flows have relevance for our understanding of fire behavior, smoke transport

and dispersion, and ecological effects such as tree mortality. Previous studies have generally focused on higher-intensity fires, including wildfires and prescribed fires (e.g., Coen et al., 2004; Clements et al., 2007; Pimont et al., 2011). In contrast, perturbation of atmospheric flows by low-intensity prescribed fires in forested landscapes has received less attention overall and remains poorly resolved in numerical modeling tools used by land managers despite the nearly 2:1 dominance of prescribed fire over wildfire in terms of area impacted, most of which is of low intensity (Melvin, 2018; Hiers et al., 2020; Melvin, 2020;

Heilman et al., 2021). This importance is further underscored by the implementation of more than 1 million prescribed fires burning more than 24 million ha in the US between 2000 and 2019 (National Interagency Fire Center, 2019). Prediction of atmospheric flows associated with low-intensity fires in forested environments is complicated by the influence of a number of interrelated factors, including near-surface meteorological conditions, local topography, forest overstory vegetation structure, and atmospheric turbulence within and above forest overstory layers (Kiefer et al., 2014; Heilman et al., 2015; Clark et al.,

2020). Since prescribed fires can impact public health and safety in nearby communities, as well as the health and safety of operational fire management personnel, improved prediction of atmospheric flows during low-intensity fires in forested environments is potentially of great benefit. Furthermore, improving the prediction of atmospheric flows during low-intensity fires may permit the refinement and operationalization of process-based modeling tools used during prescribed burning and wildland fire management operations (e.g., smoke dispersion models).

The fire-perturbed atmosphere has been studied via field experiments (e.g., Clark et al., 1999; Hiers et al., 2009; Heilman et al., 2015; Clements et al., 2019; Clark et al., 2020) and numerical modeling (e.g., Pimont et al., 2011; Hoffman et al., 2015; Kiefer et al., 2018; Kochanski et al., 2018; Linn et al., 2021). As research tools, numerical models are used to fill in gaps in our knowledge and to help answer questions that field experiments alone are unable to address (due to, for example, spatial and temporal limitations in observational data coverage, degrees of freedom that are difficult to control, and limits on repeatability).

Broadly speaking, all atmospheric numerical models solve a set of partial differential equations derived from the Navier-Stokes equations for conservation of momentum, mass, and thermodynamic energy, and parameterize physical processes too small to be explicitly resolved by the model (i.e., subgrid-scale processes). Mesoscale models are a class of atmospheric model designed to simulate atmospheric phenomena with horizontal scales of about 2-200 km, although they are also applied to microscale phenomena, with horizontal scales less than about 2 km (e.g., Kiefer et al., 2014; Peace et al., 2016; Charney et al., 2019).

Although Coen (2018) refers to the modeling of microscale phenomena, in the context of coupled atmosphere-fire models, as either "convective-scale modeling" (scales about 100 m to less than 2 km) or "large-eddy simulation modeling" (scales of less than 1 m to about 100 m), the term "mesoscale model" is used exclusively here since the models in question were originally designed for mesoscale applications. However, caution is advised when applying mesoscale models to microscale phenomena, as turbulence and other physical processes are partially resolved on the model grid and partially parameterized;

the corresponding range of grid spacings is sometimes referred to as the terra incognita or gray zone to highlight the challenges of modeling at such scales (Wyngaard, 2004; Chow et al., 2019).



Mesoscale models are capable of simulating wildland fire impacts on atmospheric flows tens of meters to hundreds of kilometers away from the fire, if combustion by-products (e.g., heat, smoke) are properly represented in the model. Examples of such mesoscale models are the Clark coupled atmosphere-fire model (e.g., Clark et al., 1996, 2004; Coen, 2005), Active Tracer High
Resolution Atmospheric Model (ATHAM) (e.g., Trentmann et al., 2006; Luderer et al., 2006), Weather Research and Forecasting (WRF)-SFIRE (e.g., Mandel et al., 2011; Kochanski et al., 2016), Meso-Non-Hydrostatic (Meso-NH)/ForeFire (e.g., Filippi et al., 2013), Advanced Regional Prediction System (ARPS) with fire heat source (e.g., Kiefer et al., 2008, 2009, 2010), ARPS plus forest canopy submodel (ARPS-CANOPY) with fire heat source (e.g., Kiefer et al., 2014, 2015, 2016, 2018), and High-Resolution Rapid Refresh (HRRR)-Smoke (e.g., Ahmadov et al., 2017, 2021). They are positioned in between computa-
tional fluid dynamics models with combustion submodels, such as the High-Resolution Model for Strong Gradient Applications (HIGRAD)/FIRETEC (Linn et al., 2002), and global-regional models that do not include combustion heat or other byproducts, such as the Global Forecast System (GFS) (Wang et al., 2019). Mesoscale models are able to simulate a broader range of atmospheric scales than computational fluid dynamics models due to the use of fewer limiting assumptions in the derivation of model prognostic equations and the inclusion of a broader suite of physical parameterizations. However, this comes at the
expense of explicit simulation of fine-scale flows near the fireline, due to the use of grid spacing too coarse to resolve smaller-scale turbulence, the use of numerical smoothing at scales near that of the model grid spacing, and the lack of an explicit combustion model (Coen, 2018). Mesoscale models must be provided synthesized information about the combustion sensible heat source and forest overstory vegetation structure to accurately simulate mean and turbulent flows associated with wildland fires in forested environments.

The primary way that atmospheric flows are impacted by wildland fires is through heat and momentum exchanges between the fire and atmosphere (Jenkins et al., 2001; Kremens et al., 2012). Sensible heat flux has received the bulk of attention in previous studies of fire-induced atmospheric perturbations (e.g., Hiers et al., 2009; Heilman et al., 2015; Clark et al., 2020), a focus that is supported by the conclusion of Luderer et al. (2009) that latent heat flux plays a much smaller role in the development of buoyant plumes above wildland fires than sensible heat flux. One of the primary challenges encountered in
applying a mesoscale model to studies of fire-perturbed flows is the representation of the fire sensible heat source in the model. Because combustion occurs at scales too small to be resolved on the mesoscale model grid, the fire sensible heat source in the model is entirely subgrid-scale. Critically, there is no standard method for representing the sensible heat released from wildland fire combustion in a mesoscale model (Clark et al., 1996, 2004; Sun et al., 2006; Kochanski et al., 2013; Kiefer et al., 2014; Kartsios et al., 2017; Kochanski et al., 2018). This is true of two-way coupled atmosphere-fire models, such as
WRF-SFIRE, in which fire-induced changes to the simulated wind field feed back on the fire (and vice-versa), and simpler one-way coupled models, such as ARPS-CANOPY with fire heat source, in which the fire perturbs the simulated wind field, but atmospheric perturbations do not feed back on the fire. Two primary methods have been implemented previously: turbulent sensible heat flux (method 1), either in the form of a heat flux profile (method 1a, HFP; e.g., Sun et al., 2006; Mandel et al., 2011; Kiefer et al., 2014) or surface heat flux (method 1b, SHF; e.g., Trentmann et al., 2006; Sun et al., 2006; Kiefer et al.,
2009, 2010; Filippi et al., 2013; Kiefer et al., 2015, 2016, 2018); and soil temperature perturbation or "hotplate" (method 2, HP; e.g., Heilman and Fast, 1992; Kiefer et al., 2008; Filippi et al., 2013). Beyond the methods themselves, the poorly defined





nature of the model parameters used in the fire sensible heat source methods serves as motivation for exploring the sensitivity of model-observation agreement to these parameters.

In this study, ARPS and ARPS-CANOPY are utilized to simulate background and fire-perturbed atmospheric conditions dur-
ing a low-intensity operational prescribed fire in the New Jersey Pine Barrens (NJPB; Heilman et al., 2021). The prescribed fire was conducted in March 2019 as part of a US Department of Defense–Strategic Environmental Research Program (SERDP)-funded project focused on multi-scale analyses of wildland-fire combustion processes in open-canopied forests (https://www.serdp-estcp.org/Program-Areas/Resource-Conservation-and-Resiliency/Air-Quality/Fire-Emissions/RC-2641, accessed 30 Sep 2021). This study is one part of a broader effort within the SERDP project to explore how atmospheric dynamics, including
ambient, fire-induced, and forest canopy-induced turbulence regimes within and near the fire environment affect fire propagation, energy exchange, and fuel consumption. The objective of this study is to provide useful guidance for improving the representation of sensible heat released from low-intensity prescribed fires in mesoscale models, through a comparison of model simulations with the combustion sensible heat source methods outlined above and a suite of observations collected during the fire. Although this study focuses exclusively on low-intensity fires, the study findings may have relevance to mesoscale
model simulations of higher-intensity fires, including wildfires. The availability of an extensive observational dataset from a low-intensity prescribed fire, and the previous application of ARPS-CANOPY to low-intensity fires, motivates this focus.

The remainder of this manuscript is organized as follows: the prescribed fire field experiment is described briefly in Sect. 2; an overview of mesoscale model fire sensible heat source methods is provided in Sect. 3; the numerical experiment methodology is discussed in Sect. 4, including descriptions of the model (4.1), model configuration and experiment design (4.2), and
analysis methodology (4.3); results and discussion are presented in Sect. 5, including details of the model assessment (5.1) and model sensitivity study (5.2) phases; and the paper is concluded in Sect. 6.

## 2  Field Experiment

The prescribed fire was conducted on 13 March 2019 in an 11.2-ha burn unit located in the NJPB at the Silas Little Experimental Forest (39.9156° N, 74.5956° W) in south-central New Jersey, USA (Fig. 1) (Heilman et al., 2021). The burn unit was
situated in a mixed oak - pine stand, with chestnut (*Q. prinus* L.), black (*Q. velutina* Lam.), white (*Q. alba* L.), and scarlet (*Q. coccinea* Münchh.) oaks, and shortleaf (*P. echinata* Mill.) and pitch (*P. rigida* Mill.) pines forming the forest overstory. Dominant trees in the area were approximately 105 years old, and the maximum height of the overstory vegetation was ∼20 m. Basal area was approximately 15.5 $\mathrm{m^2ha^{-1}}$, with oak trees and saplings and pine trees and saplings accounting for 62% and 38% of the total, respectively (Clark et al., 2018). Understory vegetation consisted of shrubs, primarily huckleberry (*Gay-*
*lussacia* spp.) and blueberry (*Vaccinium* spp.). Bear (*Q. ilicifolia* Wangenh.) and blackjack (*Q. marilandica* Münchh.) oak, sedges (*Carex pensylvanica* Lam.), and mosses were also present. The litter layer consisted of mixed fine litter of oaks, pines and understory vegetation, fine stems, and reproductive material, primarily pine cones.

Instrumentation consisted of a network of "fire-tracker" and multi-spectral sensors to measure flame arrival times, intensity, and radiative heat flux; and six instrumented flux towers (Fig. 1). Fuel loading, moisture contents, and consumption were es-

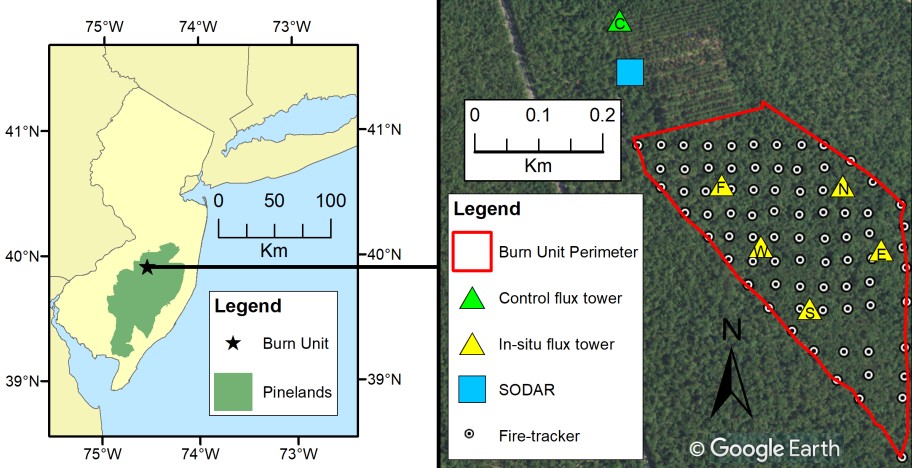

**Figure 1.** Location of the burn unit within the New Jersey Pine Barrens (NJPB) and portions of the northeastern US (left panel), and the locations of instrumentation deployed during the 13 March 2019 NJPB prescribed fire experiment (right panel); see legend in right panel for an explanation of instrumentation symbols.

timated from a combination of LiDAR and harvest measurements. To spatially characterize the spread of the prescribed fire through the burn unit, an array of 68 fire-tracker sensors was installed at ground level throughout the plot in a grid with approximately 40-m spacing between sensors (Fig. 1). The sensors consisted of 1.5-mm diameter K-type thermocouples attached to Arduino Feather data loggers (2796, Adafruit, New York, NY, USA) with thermocouple amplifiers (269, Adafruit) and GPS antennas (746, Adafruit) to time stamp and provide location for each sensor. The fire-trackers were buried such that the tip of

the thermocouple protruded through the surface fuels. All fire-tracker data were logged at a frequency of 2 Hz.

The flux towers consisted of a control tower ($TWR_C$), located approximately 180 m north-northwest of the northern burn unit perimeter, to characterize the ambient wind velocities and temperatures during the experiment, and five in-situ towers (east: $TWR_E$, flux: $TWR_F$, south: $TWR_S$, north: $TWR_N$, and west: $TWR_W$), to measure turbulent fluctuations and sensible (i.e., convective) heat flux at multiple heights above the flame fronts. On the flux towers, sonic anemometers (Model 81000V,

R. M. Young, Inc., Traverse City, MI, USA) were mounted at 3, 10, and 19 m above ground level (AGL) (hereafter, referred to as the lower, middle, and upper tower levels), and aligned in the true-north direction to measure variations in the three-dimensional wind velocity components (west-east, south-north, and vertical) and air temperature; $TWR_S$ was instrumented with sonic anemometers at the lower and upper tower levels only. Data were recorded at 10 Hz using Campbell Scientific CR3000 data loggers (Campbell Scientific, Inc., Logan, UT, USA). Additional temperature measurements were made using

thermocouples (Omega SSRTC-GG-K-36, Omega Engineering, Inc., Stamford, CT, USA) mounted at 0.25, 0.5, 1.0, 2.5, 5.0, 10.0, and 15.0 m AGL (all towers except $TWR_S$, where instruments were mounted at 0.5 m AGL and every meter from 1-10 m AGL). Also, a SOnic Detection And Ranging (SODAR) wind profiler (Remtech PA0, The Villages, FL, USA) was deployed about 80 m south of $TWR_C$ to characterize planetary boundary layer wind speed and direction up to about 400 m AGL.





At 14:57 Eastern Daylight Time (EDT) on 13 March 2019, personnel from the New Jersey Forest Fire Service ignited a
line fire using drip torches along the westernmost boundary of the burn unit, proceeding from south to north, with the first
detection by the fire-tracker array at 14:59 EDT. The line fire was allowed to spread with the wind (i.e., a head fire) from
the southwestern boundary through the burn unit and beneath the in-situ towers in a generally northeastward direction until
reaching the eastern boundary at approximately 16:30 EDT. Burning was confined to the surface and understory fuels; very little
overstory vegetation was burned. Active burning was completed at about 16:45 EDT, with smoldering continuing thereafter,
and the final detection by the fire-tracker array was at 17:38 EDT.

## 3  Fire sensible heat source methods

As stated in Sect. 1, there are two primary ways in which the fire sensible heat source has been represented in previous
mesoscale modeling studies: turbulent sensible heat flux (method 1a, HFP; method 1b, SHF), and hotplate (method 2, HP).
The choice of method determines to what degree fire-specific model parameterizations are used to represent vertical sensible
heat transport from the fire by unresolved fire-induced turbulent eddies, and to what degree the mesoscale model's native (i.e.,
non-fire condition) land-surface and subgrid-scale turbulence parameterizations are relied on to perform this function. From
HFP to SHF to HP, progressively more of the model's native parameterizations are engaged to represent vertical sensible heat
transport from the fire.

With the profile form of the turbulent sensible heat flux approach (method 1a, HFP), an exponential decay of turbulent
sensible heat flux ($H$) with height ($z$) is prescribed as a function of a surface value ($H_S$) and an extinction coefficient ($K_E$),

$$H(z) = H_S e^{-K_E z} \tag{1}$$

Equation (1) first appeared in Sun et al. (2006), and is based on Beer's law (Pfeiffer and Liebhafsky, 1959). At the height
above the ground where the fire-induced subgrid-scale turbulent sensible heat flux becomes negligible, the turbulent sensible
heat flux is computed by the model via small-eddy theory (heat flux is proportional to the local vertical gradient of potential
temperature; Stull, 1988), as is standard in subgrid-scale turbulence parameterizations in mesoscale models (e.g., ARPS, WRF;
Xue et al., 2000; Powers et al., 2017),

$$H = \overline{w'\theta'} = -K \frac{\partial \theta}{\partial z} \tag{2}$$

where $w$ and $\theta$ are vertical velocity and potential temperature, respectively, $K$ is eddy diffusivity, and $()'$ and $\overline{()}$ indicate
perturbations and grid-volume averages, respectively. In ARPS-CANOPY, $K$ is parameterized as a function of subgrid-scale
turbulent kinetic energy (TKE) and a turbulent length scale, the latter a function of grid spacing and atmospheric stability,
and separate $K$ values are computed for horizontal and vertical turbulent mixing (Xue et al., 2000; Kiefer et al., 2013). With
the surface flux form of the turbulent sensible heat flux approach (method 1b; SHF), the combustion sensible heat flux is





implemented only at the surface level ($H_S$), and small-eddy theory is used at all grid points above the lowest atmospheric
level.

Both forms of the turbulent sensible heat flux method were examined by Sun et al. (2006) via a comparison of model
simulations to data collected during the Meteotron experiment (Benech, 1976). The authors found that depositing all of the
sensible heat released from the fire within the lowest model grid layer above the ground (i.e., SHF method) led to an unrealistic
spike in near-surface buoyancy flux within the buoyant plume above the fire. They also found that prescribing an exponentially-
decaying sensible heat flux profile with an arbitrary e-folding extinction depth of 50 m resulted in an underestimation of

near-surface buoyancy flux and vertical velocity within the buoyant plume (for reference, e-folding refers to a reduction of
heat flux to $e^{-1}$ times the surface value). They suggested that an e-folding extinction depth tied to the density of soot above
the fire might provide more realistic plume quantities, but acknowledged that the e-folding extinction depth is likely to vary
depending on a number of parameters, including fire intensity, flame height, and the environment surrounding the fire (e.g.,
atmospheric conditions, forest overstory). The poorly defined nature of the extinction coefficient (e.g., Kiefer et al., 2014;

Kartsios et al., 2017; Kochanski et al., 2018) serves as motivation for exploring the sensitivity of model-observation agreement
to this parameter (Sect. 5.2).

       Because of the previous implementation of the HP approach in mesoscale model studies of wildland fires (e.g., Heilman
and Fast, 1992; Kiefer et al., 2008), this method is also explored in this study. With this method, sensible heat released from
the fire is incorporated via a soil temperature perturbation from ambient conditions. Sensible heat is communicated to the

atmosphere via a bulk aerodynamic formula, standard in mesoscale model land-surface parameterizations, in which the surface
sensible heat flux ($H_S$) is a product of the wind speed at the lowest atmospheric level ($M_0$) and heat exchange represented by
an exchange coefficient ($C_h$) and the potential temperature difference between the lowest atmospheric level and the ground
($\theta_0 - \theta_G$) (Stull, 1988),

$$H_S = \overline{w'\theta'}_S = -C_h M_0 \left(\theta_0 - \theta_G\right) \tag{3}$$

As with the SHF approach, turbulent sensible heat flux at all vertical grid levels above the lowest atmospheric level is
computed via small-eddy theory [Eq. (2)]. It is critical to point out that outside of the burn unit, and outside of the fire sensible
heat source application period for within-burn unit grid cells, the model's native land-surface and subgrid-scale turbulence
parameterizations are employed.

## 4    Numerical Experiment Methodology

### 4.1    Model Description

For numerical simulations of the NJPB prescribed fire, ARPS and ARPS-CANOPY are utilized. ARPS is a three-dimensional,
compressible, nonhydrostatic atmospheric model with a terrain-following coordinate system (Xue et al., 2000, 2001). ARPS
has been applied across a range of spatial scales, from studies of turbulent flows, using grid spacing as fine as O(1) m (Dupont



and Brunet, 2008), to studies of mesoscale and synoptic-scale phenomena, utilizing grid spacing of O(1-10) km (e.g., Xue
et al., 2001; Parker and Johnson, 2004; Michioka and Chow, 2008). In ARPS the bulk effect of a vegetation canopy on the
atmosphere is represented within a single layer, beneath the lowest model grid point (as is standard in mesoscale models, e.g.,
WRF; Powers et al., 2017). Hereafter, this model is referred to as original ARPS to distinguish it from ARPS-CANOPY.

ARPS-CANOPY is a version of the ARPS model with a canopy submodel in which the effects of vegetation elements (e.g.,
branches and leaves) on drag, turbulence production/dissipation, radiation transfer, and the surface energy budget are accounted
for through modifications to the ARPS model equations (Kiefer et al., 2013). Such changes allow for explicit simulation of
airflow through a multi-level forest canopy. The vertical profile of plant area density (Ap), defined as the one-sided area of plant
material per unit volume, is utilized in ARPS-CANOPY to represent the bulk effects of the canopy on various atmospheric
processes (e.g., drag and radiative heating/cooling). For a full description of ARPS-CANOPY, see Kiefer et al. (2013).

A 1.5-order local turbulence parameterization with a prognostic equation for subgrid-scale TKE is utilized in original ARPS
and ARPS-CANOPY, with the addition of canopy source and sink terms in the momentum and subgrid-scale TKE equations
in ARPS-CANOPY. In all simulations, the original ARPS anisotropic turbulence option is used in which both horizontal and
vertical components of eddy viscosity and diffusivity are computed; this option is recommended when vertical grid spacing is
considerably smaller than horizontal grid spacing (Xue et al., 2000). This subgrid-scale turbulence parameterization has been
tested extensively and has been found to produce the correct vertical structure of mean variables and turbulent statistics with
and without a forest canopy (e.g., Dupont and Brunet, 2008; Kiefer et al., 2013, 2014). Furthermore, a land-surface model
based on Noilhan and Planton (1989) and Pleim and Xiu (1995), and radiation physics following Chou (1990, 1992) and Chou
and Suarez (1994) are utilized [with the addition of a canopy heat source term and with canopy shading effects on the ground
energy budget accounted for (ARPS-CANOPY only)]. Lastly, for simulations with horizontal grid spacing larger than 100 m,
a non-local turbulence parameterization based on Sun and Chang (1986) is also utilized, in addition to the local turbulence
parameterization.

## 4.2 Model Configuration and Experiment Design

A multiscale modeling strategy is utilized to capture a range of atmospheric scales of motion, from synoptic (>2000 km) to
mesoscale (2-2000 km) to microscale (<2 km). A series of one-way nested simulations are performed using original ARPS and
ARPS-CANOPY, with horizontal grid spacing ranging from 4 km in the outermost domain to 30 m in the innermost domain
(Fig. 2, Table 1). Initial and lateral boundary conditions for the outermost domain are obtained from the 12-km North American
Mesoscale (NAM) model (Rogers et al., 2009). As shown in Fig. 2, the outermost domain (D1) covers the northeastern United
States, while the innermost domain (D5) covers only the area within about 4 km of the NJPB burn unit. A distinction needs to
be made here between D1-D3, in which the original ARPS model is employed, and D4 and D5, in which ARPS-CANOPY is
applied; note that the fire parameterization is only introduced in D5. Because of this one-way nesting strategy, D1-D3 simulates
synoptic- to mesoscale processes and provides boundary conditions for D4, the domain with the canopy submodel but no fire
sensible heat source, which in turn provides boundary conditions for D5, the domain with both the canopy submodel and fire
sensible heat source.





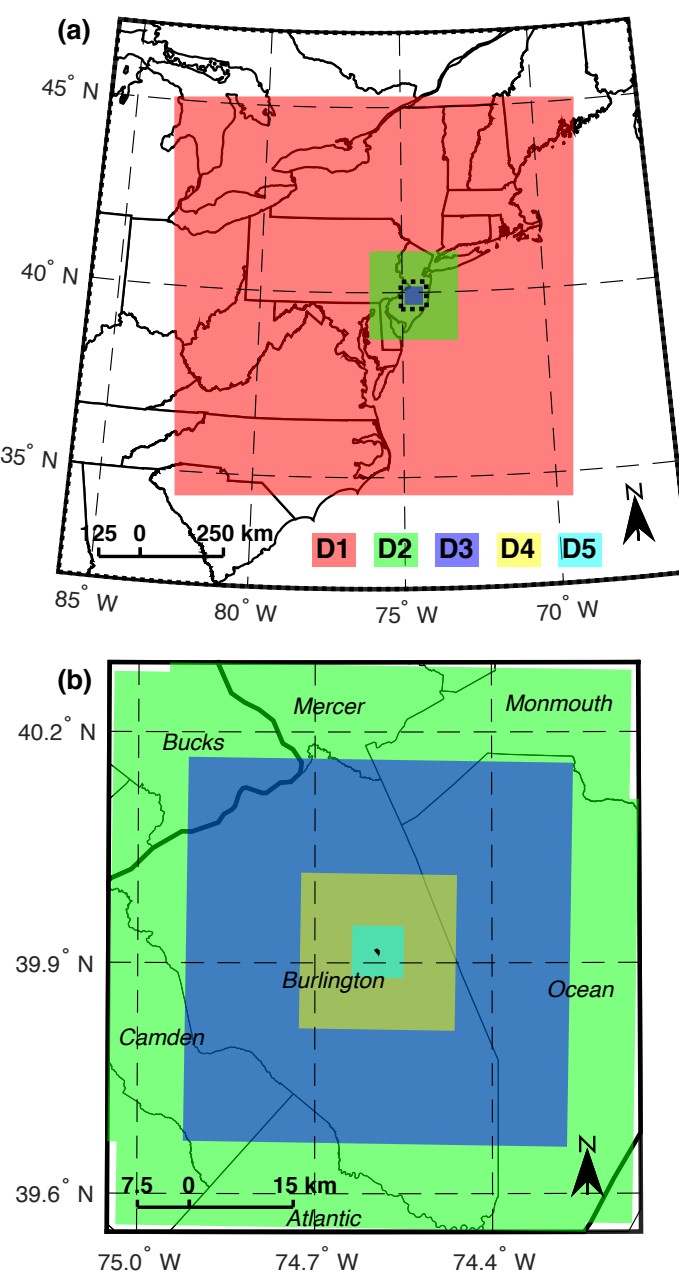

**Figure 2.** Model nesting strategy overview, at two zoom levels: (a) outer zoom with original ARPS domains D1-D3, (b) inner zoom with original ARPS domains D2-D3 and ARPS-CANOPY domains D4-D5. The dashed rectangle in (a) denotes the outline of the area displayed in (b). See Table 1 for details of each domain. Burn unit denoted by small polygon in center of domain D5. For reference, counties are labeled in (b).





**Table 1.** Nested domain summary. In the model column, "Ao" and "Ac" refer to original ARPS and ARPS-CANOPY, respectively.

| Domain | Grid size | Domain size [km] | Dx, Dy [m] | Dz min [m] | Init. time [EDT] | Duration [h] | Model |
|--------|-----------|------------------|------------|------------|------------------|--------------|-------|
| D1 | 300x300 | 1200.0 x 1200.0 | 4000 | 50.0 | 08:00 | 12 | Ao |
| D2 | 200x200 | 266.6 x 266.6 | 1333 | 25.0 | 08:00 | 12 | Ao |
| D3 | 125x125 | 55.5 x 55.5 | 444 | 12.5 | 08:00 | 12 | Ao |
| D4 | 250x250 | 22.5 x 22.5 | 90 | 2.0 | 10:00 | 10 | Ac |
| D5 | 250x250 | 7.5 x 7.5 | 30 | 2.0 | 12:00 | 6 | Ac |

The vegetation canopy is represented in ARPS-CANOPY as a height-varying Ap profile specified at each grid point. Whereas a meaningful estimation of Ap is challenging at even the scale of single tree stands, it has been demonstrated that LiDAR-derived canopy height profiles can be utilized to characterize the three-dimensional canopy structure (e.g., canopy bulk density, $kgm^{-3}$) at high horizontal and vertical resolutions on spatial scales O $(10\ km)$ (Skowronski et al., 2011; Kiefer et al., 2014). Here, we have derived Ap on grids with 90- and 30-m (horizontal) and 2-m (vertical) grid spacing from previously acquired aerial LiDAR data (Skowronski et al., 2020; Warner et al., 2020) [D5 dataset shown in Fig. 3; D4 dataset not shown]. The Ap dataset provides ARPS-CANOPY with critical information about the horizontal and vertical variability of the forest overstory vegetation (Figs. 3a-c). The resulting canopy profiles at the nearest ARPS-CANOPY grid points to the six flux towers are displayed in Figs. 3d-i.

The instrumentation deployed during the field experiment (fire-tracker array and flux towers) allows the evolution of the prescribed fire to be described in detail in the model. Data from the fire-tracker array was processed, yielding contoured maps of fire front position every 30 s (Eric Mueller, personal communication). Using GIS software, the period of fire presence within each D5 grid cell was determined. The heat source was applied in a particular burn-unit grid cell if the fire front was located anywhere within the grid cell. The heat source was applied steadily during the grid-cell fire window, with the heat source ramped up and down at the beginning and end of the fire window (total ramp-up/ramp-down period set to 10% of the fire window). A summary of the fire sensible heat source is provided in Fig. 4, beginning with a depiction of the heat source methods (and their vertical distributions) in Fig. 4a. Fireline positions at four times, corresponding to the times of peak 1-min mean turbulent sensible heat flux at towers $TWR_S$, $TWR_E$, $TWR_F$, and $TWR_N$ are depicted in Figs. 4b-e, along with the 1-min mean heat fluxes measured at all six flux towers. The time evolution of the model fire heat source area is depicted in Fig. 4f. Similar to the LiDAR-based Ap dataset, the fire-related data provides ARPS-CANOPY with important information about the spatiotemporal variability of the fire sensible heat source.

The study focuses on the D5 simulations and takes place in two phases: model assessment and model sensitivity. In the model assessment phase, analysis is limited to a control simulation ($HFP_{2.4L}$; Fig. 4a, Table 2) in which the fire sensible heat source is represented as an exponentially-decaying turbulent sensible heat flux profile (method 1a, HFP). For reference, case names

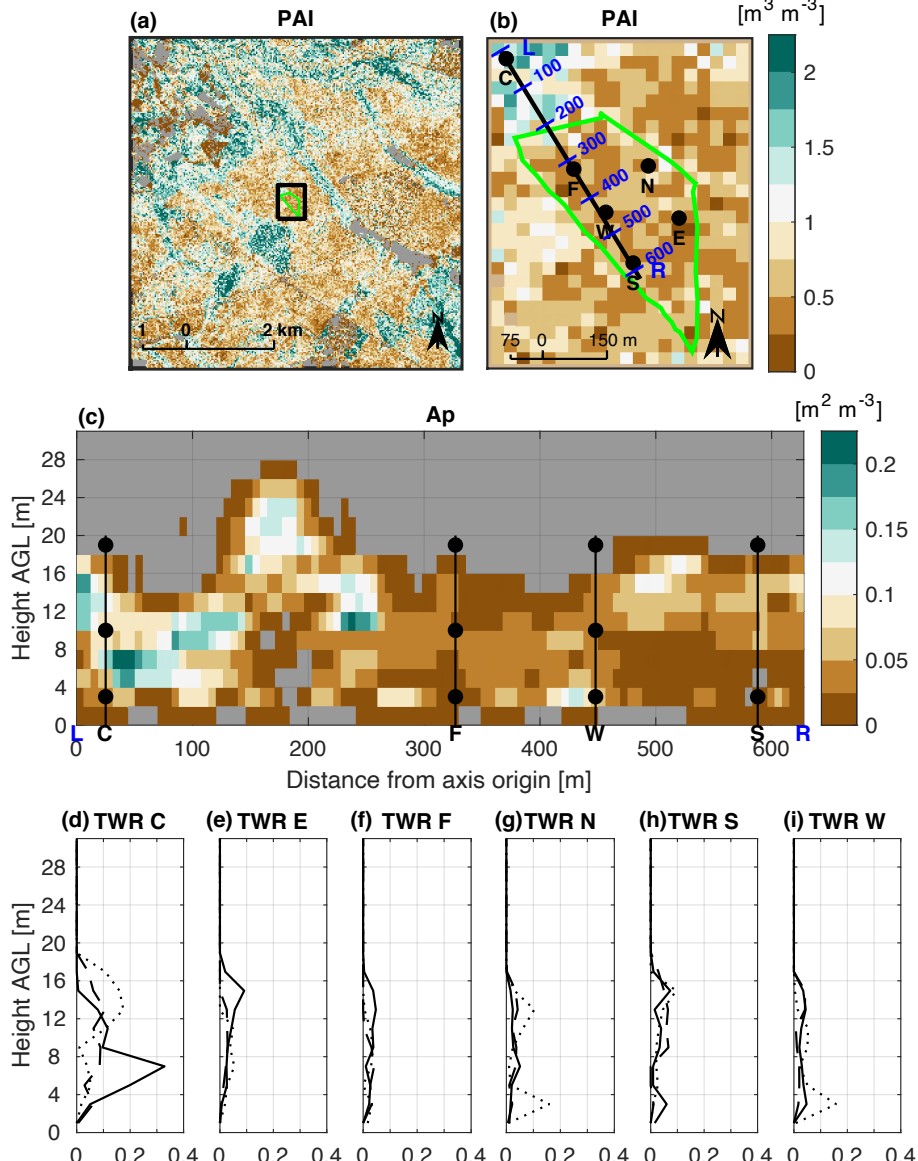

**Figure 3.** Summary of plant area density (Ap) dataset implemented in ARPS-CANOPY domain D5, derived from previously acquired aerial LiDAR data (Skowronski et al., 2020; Warner et al., 2020). The top panels depict horizontal plan views of vertically-integrated Ap [i.e., plant area index (PAI)] at two zoom levels: (a) across domain D5, and (b) within a 450-m x 450-m area centered on the burn unit; the black rectangle in (a) denotes the outline of the area displayed in (b). The middle panel (c) depicts a vertical cross section of Ap along the axis denoted by the black line in (b), with flux towers and sonic anemometer heights indicated with vertical lines and black circles, respectively. The bottom panels (d-i) depict vertical profiles of Ap $(\mathrm{m^2 m^{-3}})$ at the three model grid points nearest to each of the six flux towers; tower locations are depicted in (b). Gray shading in (a-c) denotes Ap=0.



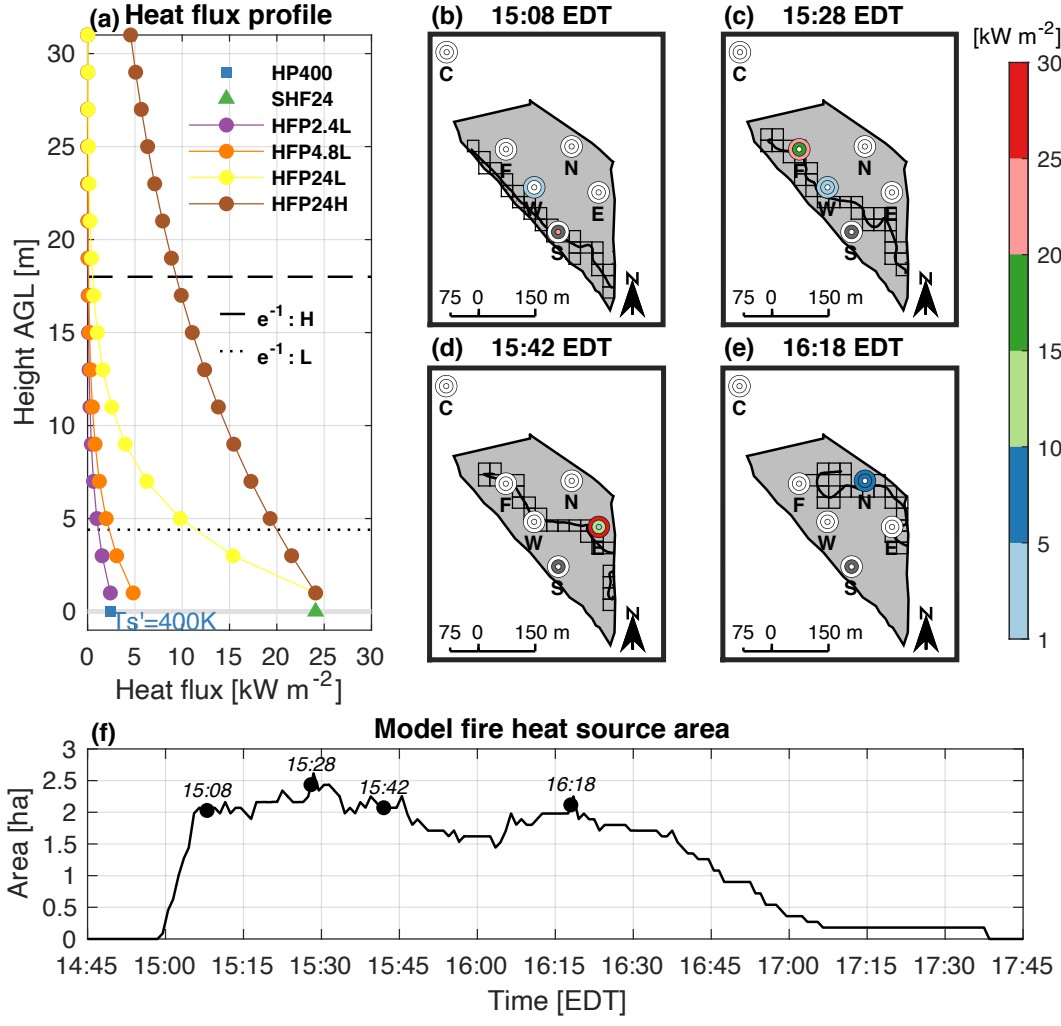

**Figure 4.** Summary of fire sensible heat source implementation in ARPS-CANOPY domain D5. Panel (a) depicts vertical profiles of ARPS-CANOPY instantaneous turbulent sensible heat flux, with the turbulent sensible heat flux method indicated by either a solid line with filled circles [method 1a, heat flux profile (HFP)] or a green triangle positioned at the surface [method 1b, surface heat flux (SHF)], and the hotplate method (method 2, HP) is indicated with a blue square positioned at the surface [with 400 K soil temperature perturbation ($T'_s$)]; panels (b-e) depict fire-tracker-derived fireline position (thick line) and tower-observed 1-min mean turbulent sensible heat flux (concentric circles), at the time of peak heat flux at (b) $TWR_S$, (c) $TWR_F$, (d) $TWR_E$, and (e) $TWR_N$, with ARPS-CANOPY grid cells subject to the sensible heat source overlaid; panel (f) depicts the time series of ARPS-CANOPY total heat source area, smoothed with a 1-min moving average filter. The dotted and dashed horizontal lines in panel (a) depict e-folding extinction depths of 4.4 m ("Low") and 18 m ("High"), respectively. For the concentric circles in panels (b)-(e): instrument height increases with distance from center, white shading is used to denote values less than $1\,\mathrm{kW\,m^{-2}}$, and dark gray shading is used to denote the absence of middle tower level measurements at $TWR_S$.





**Table 2.** Numerical experiment summary for all D5 simulations. Two primary methods are used to represent the fire sensible heat source: turbulent sensible heat flux [either a heat flux profile (method 1a, HFP) or a surface heat flux (method 1b, SHF)], and hotplate (method 2, HP). The no-fire (NF) simulation is used as the background for computing perturbations and as a reference simulation for the model sensitivity phase of the study, and simulation $HFP_{2.4L}$ is defined as the control simulation. Fire sensible heat source magnitude units are $kWm^{-2}$, except for the HP method, wherein units are K. The HFP e-folding extinction depth is either 4.4 m ("Low") or 18 m ("High").

| Simulation | Method | Magnitude | HFP e-folding depth |
|---|---|---|---|
| $HFP_{2.4L}$ | HFP | 2.4 | 4.4 (L) |
| $HFP_{4.8L}$ | HFP | 4.8 | 4.4 (L) |
| $HFP_{24L}$ | HFP | 24.0 | 4.4 (L) |
| $HFP_{24H}$ | HFP | 24.0 | 18.0 (H) |
| $SHF_{24}$ | SHF | 24.0 | — |
| $HP_{400}$ | HP | 400 | — |
| $NF_0$ | NF | 0 | — |

in this study consist of the heat source method acronym (e.g., HFP) and an alphanumeric subscript with the number denoting the heat source magnitude (e.g., 2.4 for 2.4 $kWm^{-2}$), and the letter denoting the e-folding extinction depth ["L" for low (4.4 m), "H" for high (18 m); HFP only]. The purpose of this phase is to assess the ability of ARPS-CANOPY to reproduce the

background and fire-perturbed atmosphere during the prescribed fire, as sampled by the six flux towers. Method 1a was chosen due to its successful application in a 2011 prescribed fire study in the NJPB (Kiefer et al., 2014; Heilman et al., 2015; Charney et al., 2019). Consistent with Kiefer et al. (2014), the heat flux magnitude implemented in the control simulation (2.4 $kWm^{-2}$) is 10% of the observed peak 1-min mean turbulent sensible heat flux, averaged across the in-situ towers (24 $kWm^{-2}$). This ensures that the heat released from the meters-wide fireline and measured at the flux towers (i.e., single points) is diluted before

it is implemented in each 30-m x 30-m burn-unit grid cell. An e-folding extinction depth of 4.4 m ("L" in case name subscript) is utilized, corresponding to approximately 25% of the average canopy height in the burn unit (18 m). The 4.4-m e-folding extinction depth is based on the assumption that the interception and break-down of small-scale turbulent eddies by the forest overstory vegetation will yield negligible fire-induced subgrid-scale turbulent sensible heat flux at canopy top. This e-folding extinction depth is considerably shallower than the 41-m depth used in Kiefer et al. (2014); however, the depth in that study

was chosen based on sensitivity experiments and had no physical basis. Sun et al. (2006) suggested that the e-folding extinction depth may be proportional to fire intensity: the choice of a 4.4-m e-folding extinction depth in this study is consistent with the lower-intensity nature of the 2019 NJPB prescribed fire (peak 1-min mean turbulent sensible heat flux: 26.7 $kWm^{-2}$ vs. 155.5 $kWm^{-2}$ during the 2011 NJPB fire).

In the model sensitivity phase, a series of sensitivity experiments (including the control simulation, $HFP_{2.4L}$) are conducted

in an effort to explore the sensitivity of model-observation agreement to (i) the method used to represent the fire sensible heat





source in the model [turbulent sensible heat flux (method 1a, HFP; method 1b, SHF) and soil temperature perturbation (method 2, HP)]; and (ii) HFP parameters [surface value and e-folding extinction depth] (Fig. 4a, Table 2). In the set of HFP simulations, the surface value of turbulent sensible heat flux is varied between 2.4 and 24 $\mathrm{kWm^{-2}}$; and the e-folding extinction depth (a measure of how sharply the heat flux decays with height) is varied between 4.4 and 18 m; the larger the e-folding depth, the

more gradually the heat flux decays away from the surface. For the SHF simulation ($\mathrm{SHF_{24}}$), a 24 $\mathrm{kWm^{-2}}$ surface sensible heat flux is implemented, and for the HP simulation ($\mathrm{HP_{400}}$), the soil temperature perturbation is set to 400 K, corresponding to the 68-sensor–median peak fire-tracker temperature deviation from pre-fire conditions.

## 4.3 Analysis methodology

In this study, ARPS-CANOPY simulations are assessed using sonic anemometer measurements of the background and fire-

perturbed atmosphere. Although the tower thermocouple and SODAR data have been analyzed (not shown), the sonic anemometer data are the exclusive focus of this study since unlike the former data sources, sonic anemometer measurements provide us the ability to evaluate model-simulated variances (i.e., TKE) as well as mean quantities. For observational data and model output, 1-min means are computed from instantaneous time series (10 Hz sampling frequency for observations; 1 Hz output frequency for model). For observations, quality control and assurance is performed in a four-step procedure. First, tower data

are subjected to a despiking and filtering routine to remove erroneous data and values exceeding six standard deviations from running 1-h means (Heilman et al., 2015). Second, a double-rotation tilt-correction routine is applied to the vertical wind-velocity component data to correct the raw data for systematic errors originating from errors in the physical leveling of sonic anemometers in the field (Wilczak et al., 2001). Third, means computed from fewer than 90% of possible instantaneous observations (i.e., fewer than 540/600 values) are masked. Finally, manual quality control is performed, resulting in the masking of

lower tower level temperature at $\mathrm{TWR_C}$ and $\mathrm{TWR_F}$, and upper tower level wind direction at $\mathrm{TWR_F}$.

In the model assessment phase, box-whisker plot vertical profiles, time series, vertical cross-sections, and three-dimensional surface plots of 1-min mean temperature, wind speed, wind direction, vertical velocity, and TKE, are compared between the control simulation ($\mathrm{HFP_{2.4L}}$) and observations at all six towers. The multi-dimensional approach taken in this study phase allows for a more comprehensive assessment of simulated and observed variables than point comparisons alone. The approach

provides helpful context for the assessment of differences between the model simulation and observations at the individual tower locations; for example, contoured plots can help identify spatial displacement errors (i.e., correct magnitude but incorrect location; Brown et al., 2011). As in Kiefer et al. (2014), the assessment of the control simulation in this phase is qualitative. However, the multi-dimensional nature of the analysis and greater number of flux towers in this study distinguishes this ARPS-CANOPY assessment from that of Kiefer et al. (2014).

In the model sensitivity phase, box-whisker plot vertical profiles and time series of 1-min mean temperature, wind speed, vertical velocity, and TKE, from all D5 simulations (Table 2), are compared to observations at $\mathrm{TWR_W}$. This tower is chosen for the model sensitivity analysis as it is one of three towers ($\mathrm{TWR_E}$, $\mathrm{TWR_F}$, and $\mathrm{TWR_W}$) where the control simulation was found to considerably underestimate TKE at the upper tower level (Sect. 5.1); evaluation statistics at the other towers show similar model sensitivity to that at $\mathrm{TWR_W}$ (cf. Tables 3-5 to Tables S1-S3 in supplemental material). The focus on TKE prediction is justified





by the important role that turbulence plays in affecting fire behavior (Banerjee et al., 2020) and in controlling fire effects such as tree mortality (e.g., Kavanagh et al., 2010) and smoke dispersion (e.g., Charney et al., 2019). For the model sensitivity analysis, wind direction is omitted to enable larger figure panels that can accommodate the simultaneous plotting of all seven sensitivity tests (Table 2). The evaluation of the D5 simulations in this phase is part qualitative and part quantitative. First, a qualitative evaluation of box-whisker plot vertical profiles and time series is conducted. Second, a quantitative evaluation of

the model simulations is performed using three statistics: mean difference, expressed as a percentage of the range of observed values,

$$MD = \frac{\frac{1}{n}\sum_{i=1}^{n}(M_i - O_i)}{O_{max} - O_{min}} \qquad (4)$$

root-mean-square difference, also expressed as a percentage of the range of observed values,

$$RMSD = \frac{\sqrt{\frac{1}{n}\sum_{i=1}^{n}(M_i - O_i)^2}}{O_{max} - O_{min}} \qquad (5)$$

and index of agreement (Willmott, 1981),

$$IA = 1 - \left[\frac{\sum_{i=1}^{n}(M_i - O_i)^2}{\sum_{i=1}^{n}\left(|M_i - \overline{O}| + |O_i - \overline{O}|\right)^2}\right] \qquad (6)$$

where $O$ is the observed time series, $M$ is the model-simulated time series, $n$ is the number of values, $||$ indicates the absolute value, and $max$, $min$, and $\overline{()}$ indicates the time series maximum, minimum, and mean, respectively. The normalization of MD and RMSD using the range of observed values allows for a direct comparison of statistics among variables with different

units. For reference, an IA value of 1 indicates perfect agreement between model simulation and observation, and a value of 0 indicates complete disagreement. This set of statistics allows for a quantitative evaluation of systematic (MD and IA) and random (RMSD and IA) model error, and includes statistics with varying degrees of outlier sensitivity (more sensitive: RMSD and IA, due to the use of squared differences; less sensitive: MD).

For both study phases, box-whisker plot vertical profiles and time series are constructed from 40 1-min mean values, 20

before and 20 after the time of peak turbulent sensible heat flux at each tower; model evaluation statistics are computed over the same 40-min period. The choice of a 40-min analysis period is a compromise between two goals: the time period should be long enough to include the pre- and post-fire front passage periods, and yield a large enough sample size for meaningful statistics, but short enough that the fire frontal passage period is not overly diluted. For the vertical cross-sections and three-dimensional surface plots presented in the model assessment phase, plots are constructed from the 1-min mean values corresponding to the

time of peak turbulent sensible heat flux at each tower. It is important to state that the purpose in comparing plots of 1-min mean values between the model simulations and tower observations is to judge overall model performance. There is no expectation here that the model simulation and observations will agree 100% on a minute-to-minute basis. At spatial scales on the order





of 100 m or smaller, turbulence limits the accuracy of deterministic predictions, necessitating the use of model ensembles and probabilistic verification techniques (Weigel, 2011; Coen, 2018; Chow et al., 2019). However, the computational expense of
running operational model ensembles in real (or near-real) time at such small spatiotemporal scales justifies this assessment of deterministic model predictions.

## 5 Results and Discussion

### 5.1 Model Assessment

Examination of the control simulation ($HFP_{2.4L}$) begins with an assessment of box-whisker plot vertical profiles of simulated
and observed temperature, wind speed, wind direction, vertical velocity, and TKE, at each tower (Fig. 5). This assessment centers on three questions regarding model – observation agreement: (1) Does the model capture the vertical variability sampled by the lower, middle, and upper tower level sonic anemometers? (2) Does the model capture the range of values observed during the 40-min analysis window? (3) Which variable(s) does the model simulate with highest fidelity to observations and which variable(s) does it simulate with lowest fidelity? Regarding (1), the model is able to generally reproduce the observed vertical
variability for all five variables, at all six towers. The model depicts a statically unstable atmosphere above the fire with light southerly winds increasing in magnitude with height, and increasing vertical velocity magnitude and turbulence intensity with height through the overstory vegetation layer. Regarding (2), while the model generally captures the range of values during the 40-min window, there is a tendency to underestimate values on the right side of the box-whisker plot. Consistent with observations, the range of values simulated during the 40-min window generally decreases with height for temperature and
wind direction, and increases with height for wind speed, vertical velocity, and TKE. Regarding (3), of the five variables, wind speed and vertical velocity appear to be simulated with highest fidelity, and temperature and TKE simulated with lowest fidelity. The model captures the vertical wind speed profile shape, including the positive wind shear between the middle and upper tower level sonic anemometers, and captures the increase in vertical velocity magnitude with height. However, the model exhibits a negative temperature bias, even outside the burn unit at $TWR_C$, and the model considerably underestimates TKE at
the upper tower level, in particular at towers $TWR_E$, $TWR_F$, and $TWR_W$.

Proceeding to the time series of simulated and observed variables (Fig. 6), the focus of the analysis shifts to an assessment of temporal variability and overall agreement in the timing of fire-induced variations (relative to the time of peak 1-min mean observed turbulent sensible heat flux). In examining Fig. 6, it is critical to keep in mind that the timing and duration of enhanced values of sensible heat flux ("fire window") in each ARPS-CANOPY grid cell is informed by the fire-tracker data, not the flux
tower data. The fire window within each model grid cell is generally longer than the fire window observed at the towers, and the model and tower-observed fire windows may begin and end at different times. Examining Fig. 6, the model is found to generally capture the temporal variability and range of wind speed, wind direction, and to a lesser degree vertical velocity, but not temperature and TKE. For the latter two variables, the magnitudes are considerably underestimated, and the timing of the fire-induced peak in simulated and observed variables does not generally agree. It is worth noting that the relative timing of the
peaks in *observed* turbulent sensible heat flux, temperature, and TKE differ from tower to tower. For example, at $TWR_N$ peak



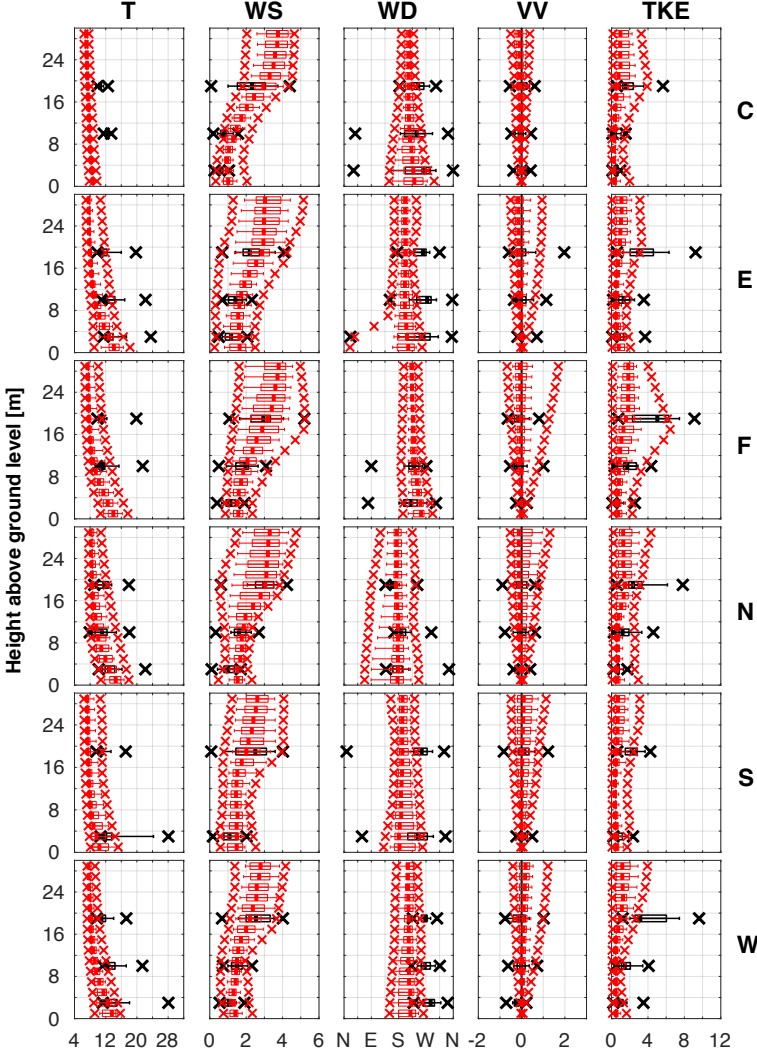

**Figure 5.** Box-whisker plot vertical profiles of 1-min mean temperature (T; $^\circ$C), wind speed (WS; ms$^{-1}$), wind direction (WD; deg), vertical velocity (VV; ms$^{-1}$), and turbulent kinetic energy (TKE; m$^2$s$^{-2}$), for control simulation (HFP$_{2.4L}$), at all towers (labeled along right axis). Box-whisker plots are constructed from 40 1-min mean values, $\pm$20 min from the time of peak 1-min mean observed heat flux, except for TWR$_C$ wherein 15:20 EDT is chosen arbitrarily: TWR$_E$, 15:42 EDT; TWR$_F$, 15:28 EDT; TWR$_N$, 16:20 EDT; TWR$_S$, 15:08 EDT; TWR$_W$, 15:38 EDT. For each box-whisker plot, the thick line denotes the median, the boxes extend outward to the 25$^{th}$ and 75$^{th}$ percentiles, the whiskers extend outward to the 10$^{th}$ and 90$^{th}$ percentiles, and the "x" symbols indicate the minimum and maximum values. Tower observations and model-simulated values are indicated by black and red colors, respectively. Vertical black line in VV panels corresponds to zero vertical velocity. The x-axis in the WD column is labeled as follows: N (0/360), E (90), S (180), W(270). Manual quality control of tower observations results in the exclusion of three data points: lower tower level T at TWR$_C$ and TWR$_F$, and upper tower level WD at TWR$_F$.





temperature occurs approximately 2.5 min *after* the time of peak turbulent sensible heat flux (vertical black line), whereas peak TKE occurs about 10 min *before* the time of peak turbulent sensible heat flux. At TWR$_W$ peak temperature and TKE occur simultaneously about 2.5 min *before* the time of peak turbulent sensible heat flux (vertical black line). The differences in timing of peaks among different flux towers and different variables underscores some of the challenges of deterministic model

assessment at the microscale.

To provide context for the assessment of ARPS-CANOPY at the flux towers, vertical cross-sections are presented in Fig. 7, along the same northwest–southeast oriented axis used earlier for the Ap vertical cross-section (Fig. 3b-c), and 3D surface plots are presented in Fig. 8. These multi-dimensional plots are intended to help identify displacement errors that can result when the model correctly simulates the magnitude, but not the location, of fire-induced perturbations. Errors in both magnitude and

location complicate the model assessment process (see Fig. 6.3 in Brown et al., 2011). Analysis begins with the vertical cross-sections in Fig. 7: from northwest to southeast, the approximately 600-m–long cross-section axis intersects TWR$_C$, TWR$_F$, TWR$_W$, and TWR$_S$. Examining Fig. 7 as a whole, evidence of displacement error is found in most panels, but in particular the temperature, wind speed, and TKE panels. An example of spatial displacement error is seen in the left panel of the second row (see wind speed at TWR$_C$): observed wind speed is less than 0.5 ms$^{-1}$ and simulated wind speed is 1-2 ms$^{-1}$; however, values

around 0.5 ms$^{-1}$ are simulated about 150 m southeast of TWR$_C$ (a distance of about five grid cells). As a further example, in the bottom center and bottom right panels (see TKE at TWR$_F$ and TWR$_W$), maxima are simulated by the model adjacent to but not colocated with the towers. Although the simulated maxima are still less than what is observed at the towers, the values are 1-2 m$^2$s$^{-2}$ higher than the simulated values interpolated to the tower locations (Fig. 5). One-dimensional analyses like vertical profiles and time series (Figs. 5-6), although certainly useful for model assessment purposes, fail to fully characterize

model-observation agreement due to their inherent inability to detect spatial displacement error.

Assessment of of the control simulation concludes with an examination of horizontal slices of 1-min mean variables at the time of peak heat flux at TWR$_W$ (15:38 EDT), rendered in 3D space (Fig. 8). Comparing the fireline position, derived from the fire-tracker array, to the positions of simulated variable maxima and minima, enhanced values of temperature (Fig. 8a), wind speed (Fig. 8b), and to a lesser degree, TKE (Fig. 8e), are found downwind (northeast) of the fireline at the 3 and 9 m model

grid levels. For the other variables, and at the 19 m model grid level for all variables, it is difficult to discern any relationship between the simulated variables and the fireline. Linear structures in the wind speed (Fig. 8b), vertical velocity (Fig. 8d), and TKE (Fig. 8e) panels, most noticeable at 19 m AGL, are suggestive of planetary boundary layer structures aligned with the mean wind (oriented southwest-northeast). Regarding TKE specifically, Fig. 8e reveals 1-min mean simulated TKE values of 5-7 m$^2$s$^{-2}$ at 19 m AGL within and adjacent to the burn unit at a moment when observed upper tower level TKE values at the

five in-situ towers are between 1.8 and 9.2 m$^2$s$^{-2}$. Although this comparison is encouraging, it is important to note that this does not necessarily imply that the model is correctly capturing the processes yielding the observed TKE values at the towers. What Figs. 5-8 do suggest is that the model is reproducing the overall background and fire-perturbed atmosphere depicted by the tower observations. The shortcoming in accurately simulating upper tower level TKE values at TWR$_E$, TWR$_F$, and TWR$_W$ motivates the model sensitivity tests that follow.



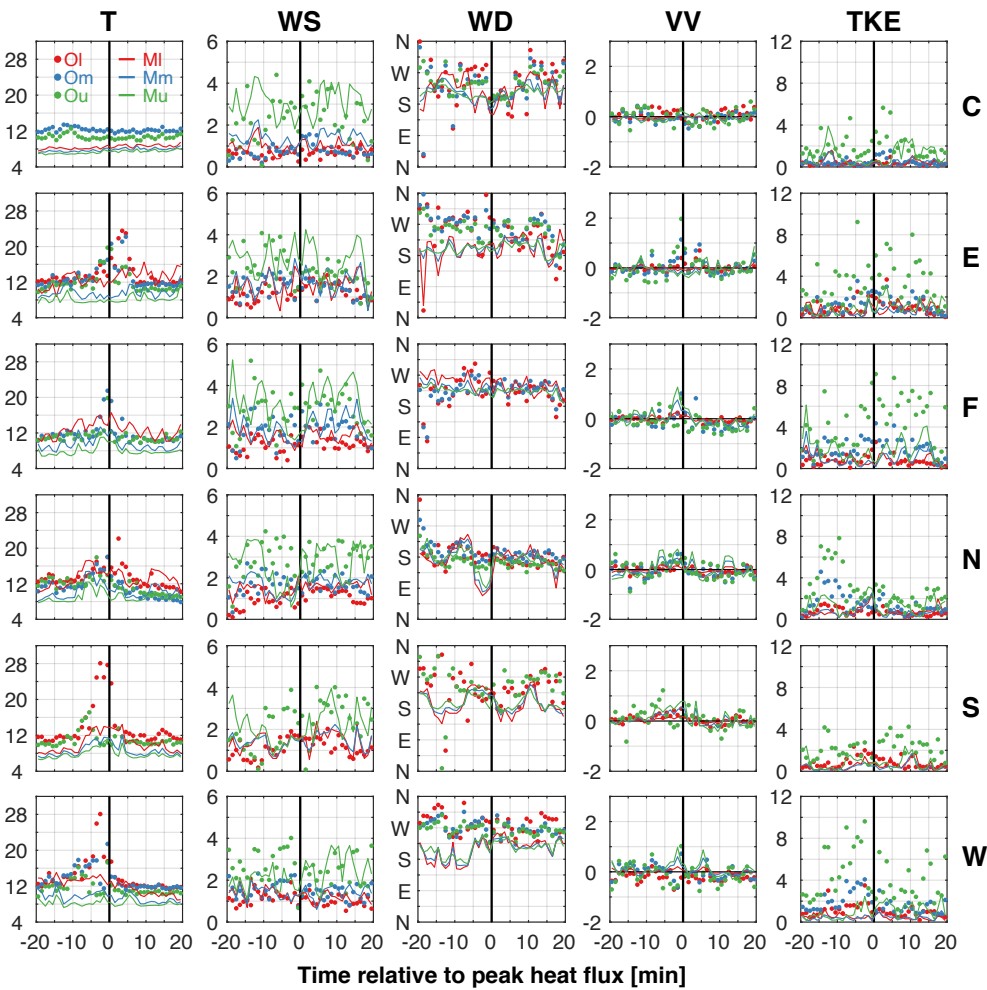

**Figure 6.** Time series of 1-min mean temperature (T; °C), wind speed (WS; $ms^{-1}$), wind direction (WD; deg), vertical velocity (VV; $ms^{-1}$), and turbulent kinetic energy (TKE; $m^2 s^{-2}$), for control simulation (HFP$_{2.4L}$), at all towers (labeled along right axis). Time series are constructed from 40 1-min mean values, ±20 min from the time of peak 1-min mean observed heat flux, except for TWR$_C$ wherein 15:20 EDT is chosen arbitrarily: TWR$_E$, 15:42 EDT; TWR$_F$, 15:28 EDT; TWR$_N$, 16:20 EDT; TWR$_S$, 15:08 EDT; TWR$_W$, 15:38 EDT. Observed (O) and model (M)-simulated values (vertically interpolated to sonic anemometer levels) are indicated by filled circles and lines, respectively, and lower (l), middle (m), and upper (u) tower levels are indicated with red, blue, and green colors, respectively (see legend in top-left panel). Vertical black line in each panel indicates time of peak 1-min mean observed heat flux; horizontal black line in VV panels corresponds to zero vertical velocity. The y-axis in the WD column is labeled as follows: N (0/360), E (90), S (180), W(270). Manual quality control of tower observations results in the exclusion of three data points: lower tower level T at TWR$_C$ and TWR$_F$, and upper tower level WD at TWR$_F$.



**Figure 7.** Vertical cross-sections of 1-min mean temperature (T; °C), wind speed (WS; ms$^{-1}$), wind direction (WD; deg), vertical velocity (VV; ms$^{-1}$), and turbulent kinetic energy (TKE; m$^2$s$^{-2}$), for control simulation (HFP$_{2.4L}$), along the axis shown in Fig. 3b. Times displayed correspond (from left to right) to the time of peak turbulent sensible heat flux observed at the three in-situ towers intersected by the cross-section axis, TWR$_S$ (15:08 EDT), TWR$_F$ (15:28 EDT), and TWR$_W$ (15:38 EDT); corresponding tower initial highlighted with magenta font. Tower observations and model-simulated values are indicated by filled circles and shading, respectively. Color bar in WD row is labeled as follows: N (0/360), E (90), S (180), W(270). Manual quality control of tower observations results in the exclusion of three data points: lower tower level T at TWR$_C$ and TWR$_F$, and upper tower level WD at TWR$_F$.



**Figure 8.** Horizontal slices of 1-min mean temperature (T; °C), wind speed (WS; $\mathrm{ms}^{-1}$), wind direction (WD; deg), vertical velocity (VV; $\mathrm{ms}^{-1}$), and turbulent kinetic energy (TKE; $\mathrm{m}^2\mathrm{s}^{-2}$), for control simulation (HFP$_{2.4L}$), at 3, 9, and 19 m model grid levels. All panels correspond to the time of peak turbulent sensible heat flux observed at TWR$_W$ (15:38 EDT). Observed and model-simulated values are indicated by filled circles and shading, respectively. Magenta line inside burn unit is the fireline position derived from the fire-tracker array. See bottom-right reference panel (f) for tower names. Color bar in WD panel (c) is labeled as follows: N (0/360), E (90), S (180), W(270). Spacing between x- and y-axis tick marks is 90 m. Manual quality control of tower observations results in the exclusion of three data points: lower tower level T at TWR$_C$ and TWR$_F$, and upper tower level WD at TWR$_F$ (denoted by black circles).





## 5.2  Model Sensitivity

In this study phase, box-whisker plot vertical profiles and time series of simulated temperature, wind speed, vertical velocity, and TKE from all seven D5 simulations are examined at $TWR_W$ (Figs. 9-10), along with model evaluation statistics (Tables 3-5). Recall from Sect. 4.3 that this tower is chosen for the model sensitivity analysis as it is one of three towers ($TWR_E$, $TWR_F$, and $TWR_W$) where the control simulation ($HFP_{2.4L}$) was found to considerably underestimate TKE at the upper tower

level (Figs 5-8). The reader is reminded that this evaluation of simulations with different fire sensible heat source methods is intended to ultimately provide guidance for improving the representation of the sensible heat released from low-intensity prescribed fires in mesoscale models. The relevance of temperature, wind speed, and vertical velocity to predictions of, for example, smoke dispersion and fire behavior, motivates the examination of these variables in addition to TKE.

Beginning the model sensitivity evaluation with box-whisker plot vertical profiles of temperature (Fig. 9a), all simulations

correctly depict decreasing median and range of temperature, with height. However, a broad range of lower tower level temperatures (and consequently, buoyancy) during the 40-m analysis period is found among the simulations. The $NF_0$, $HFP_{2.4L}$, $HFP_{4.8L}$, and $HP_{400}$ simulations yield temperatures in relatively narrow ranges of 8.4-10.8, 9.2-15.8, 8.9-20.2, and 9.2-19.4 °C, respectively, whereas the $HFP_{24L}$ and $SHF_{24}$ simulations yield temperatures in the much broader ranges of 9.3-49.4 °C and 9.8-63.8 °C, respectively. The simulation that most closely agrees with the observed 11.3 - 28.1 °C range at the lower

tower level is $HFP_{24H}$, with a range of 9.1-24.2 °C. Proceeding to wind speed (Fig. 9b), the observed vertical variation in wind speed median and range is reproduced in all simulations, with the important exception of select simulations at the lower tower level. In three of the simulations ($HFP_{24L}$, $HFP_{24H}$, and $SHF_{24}$), lower tower level wind speeds are overestimated by 1-2 $ms^{-1}$. It appears that excessive near-surface wind speed results when model simulations are forced by a turbulent sensible heat flux corresponding to 100% of the peak 1-min mean observed heat flux averaged across the flux towers inside the burn unit,

regardless of how the heat flux is distributed vertically.

Proceeding to vertical velocity (Fig. 9c), it appears that all simulations capture the observed increase in vertical velocity median and range with height, including the simulations with no or weak sensible heat sources ($NF_0$, $HFP_{2.4L}$, $HFP_{4.8L}$, and $HP_{400}$), yielding maximum positive values of about 0.2, 0.7, and 1 $ms^{-1}$ at the lower, middle, and upper tower levels, respectively. Although the model correctly simulates median vertical velocity of about 0 and -0.1 $ms^{-1}$ at the middle and upper tower

levels, respectively, none of the simulations capture the median vertical velocity of about -0.25 $ms^{-1}$ at the lower tower level, or the maximum negative values of -0.6– -0.75 $ms^{-1}$ at all three levels. The inability of the model to adequately simulate negative vertical velocity is also apparent in Fig. 5, particularly at $TWR_N$ and $TWR_S$, in addition to the aforementioned model deficiency at $TWR_W$. The negative vertical velocity observed at these towers may be evidence of downdrafts directly behind the fire front, identified previously in Heilman et al. (2015).

Finally, examination of TKE (Fig. 9d) reveals that although the observed vertical variation of TKE median and range with height is generally reproduced among the seven simulations, there is considerable sensitivity of TKE to the choice of fire sensible heat source method and heat flux profile parameters, especially at the upper tower level. At this level, the peak simulated TKE is $\sim 6$ $m^2s^{-2}$ in $HFP_{24L}$, $HFP_{24H}$, and $SHF_{24}$, but less than 3 $m^2s^{-2}$ in the other simulations. Although none



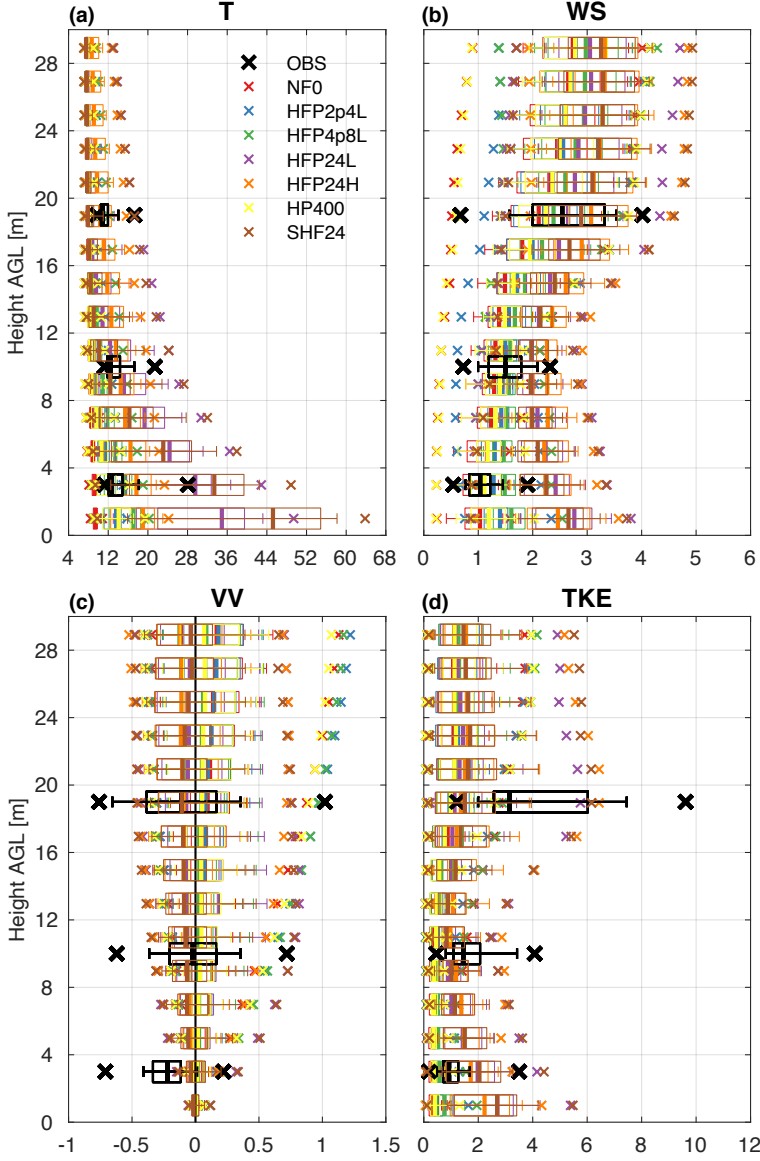

**Figure 9.** Box-whisker plot vertical profiles of 1-min mean temperature (T; °C), wind speed (WS; ms$^{-1}$), vertical velocity (VV; ms$^{-1}$), and turbulent kinetic energy (TKE; m$^2$s$^{-2}$), for all D5 simulations, at TWR$_W$. Box-whisker plots are constructed from 40 1-min mean values, ±20 min from the time of peak 1-min mean observed heat flux: 15:38 EDT. For each box-whisker plot, the thick line denotes the median, the boxes extend outward to the 25$^{th}$ and 75$^{th}$ percentiles, the whiskers extend outward to the 10$^{th}$ and 90$^{th}$ percentiles, and the "x" symbols indicate the minimum and maximum values. See legend in (a) for symbol color correspondence to tower observations and model simulations. Vertical black line in VV panel (c) corresponds to zero vertical velocity.





of the simulations achieve the observed peak of $\sim 9.5 \ \mathrm{m^2 s^{-2}}$, and the peak simulated values correspond approximately to

the 75$^{\text{th}}$ percentile of observed values, it is clear that the simulations with 100% of the peak 1-min mean observed heat flux implemented best capture the TKE near or just above the top of the forest overstory. However, each of these three simulations (HFP$_{24L}$, HFP$_{24H}$, and SHF$_{24}$) overestimate TKE at the lower tower level, with the HFP$_{24H}$ simulation closest to the observed values.

The model sensitivity exercise concludes with a qualitative evaluation of 1-min mean time series (Fig. 10) and a corre-

sponding quantitative evaluation of model verification statistics: mean difference [Eq. (4)], root mean square difference [Eq. (5)], and index of agreement [Eq. (6)] (Tables 3-5). Beginning with the time series of temperature (Fig. 10), three aspects of model-observation agreement stand out: first, all simulations exhibit a cold bias that increases with height (see especially the post-fire period); second, the timing and duration of the temperature peak varies between the simulations, with the peak occurring earliest in simulations with the most intense heat source (e.g., HFP$_{24L}$); third, the two HFP simulations with 100%

of the peak 1-min mean observed heat flux concentrated at or near the surface (HFP$_{24L}$ and SHF$_{24}$) exhibit pronounced and persistent warm biases at the lower tower level. Proceeding to the wind speed panels in Fig. 10, the sensitivity to the sensible heat source method and parameters is shown to increase from the top of the tower to the bottom. At the upper tower level, all simulations depict similar temporal variability and magnitudes. However, at the middle and especially at the lower tower level, the seven simulations are found to cluster in two groups: the three simulations with 100% of the peak 1-min mean observed

turbulent sensible heat flux implemented (HFP$_{24L}$, HFP$_{24H}$, and SHF$_{24}$) form a "strong" wind speed group (wind speed$\sim$1.5-3 ms$^{-1}$), with the other simulations clustered in a "weak" wind speed group (wind speed$\sim$0.5-2.5 ms$^{-1}$). It is worth noting that despite differences in the vertical distribution of the sensible heat source in the "strong" wind speed group, wind speed differs little between the three simulations.

Examining vertical velocity in Fig. 10, only modest sensitivity to the sensible heat source method and parameters is noted.

The most noticeable aspect of the vertical velocity panels is the general lack of negative values at the lower tower level. Regardless of sensible heat source implementation, none of the simulations capture the magnitude of negative vertical velocity observed at TWR$_{\text{W}}$. This shortcoming may reflect the inability of the model to fully resolve, with 30-m horizontal grid spacing and 2-m vertical grid spacing, turbulent eddies with spacial scales on the order of meters or less. Finally, the TKE panels depict a similar relationship between the simulations as for wind speed. With decreasing height, the simulations with 100% of the

peak 1-min mean observed turbulent sensible heat flux implemented (HFP$_{24L}$, HFP$_{24H}$, and SHF$_{24}$) increasingly cluster in one group, and all other simulations cluster in a second group. The time series of wind speed and TKE at the lower tower level closely correspond, with the first simulation group exhibiting persistently large TKE. It is important to note a critical model shortcoming: at the upper tower level, observed (simulated) TKE is persistently (briefly) large, relative to the background state; at the lower tower level, the opposite is found. Thus, simulations with 100% of the peak 1-min mean observed turbulent

sensible heat flux implemented (HFP$_{24L}$, HFP$_{24H}$, and SHF$_{24}$) yield peak 1-min mean TKE values comparable to the observed values, but the persistence (or lack thereof) is not well captured. When longer time averages are computed, the simulated TKE values are generally largest at the lower tower level and smallest at the upper tower level.



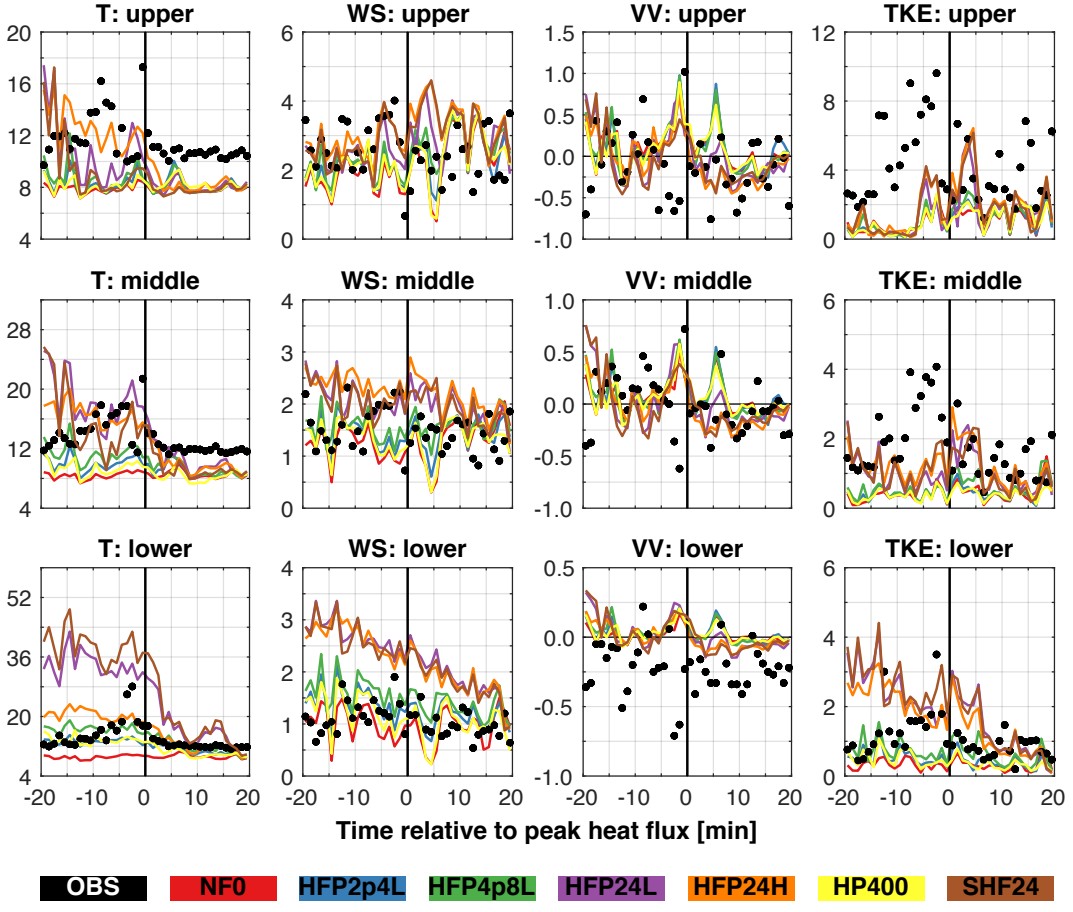

**Figure 10.** Time series of 1-min mean temperature (T; °C), wind speed (WS; ms$^{-1}$), vertical velocity (VV; ms$^{-1}$), and turbulent kinetic energy (TKE; m$^2$s$^{-2}$), for all D5 simulations, at TWR$_W$. Time series are constructed from 40 1-min mean values, ±20 min from the time of peak 1-min mean observed heat flux: 15:38 EDT. Observed and model-simulated values (vertically interpolated to sonic anemometer levels) are indicated by filled circles and lines, respectively. Vertical black line in each panel indicates time of peak 1-min mean observed heat flux; horizontal black line in VV panels corresponds to zero vertical velocity. Note the difference in y-axis limits between rows. See legend at bottom for color correspondence to tower observations and model simulations.

The model sensitivity exercise concludes with a quantitative evaluation of the ARPS-CANOPY simulations using the model verification statistics presented in Tables 3-5. Examining MD and RMSD first, the largest percent differences are found for temperature and wind speed, with the largest positive temperature differences coinciding with the largest positive wind speed differences (cf. Tables 3-4 and Fig. 9). Lower tower level vertical velocity is consistently overestimated by ARPS-CANOPY, and upper tower level TKE is consistently underestimated. Examining IA second, the lowest overall values are found for the lower and middle tower levels, with IA as low as about 0.25 (recall from Sect. 4.3 that IA values of 0 and 1 correspond to complete disagreement and complete agreement between model and observations, respectively). Considering simultaneously



**Table 3.** Mean difference (simulation – observation) for 1-min mean temperature (T; °C), wind speed (WS; ms$^{-1}$), vertical velocity (VV; ms$^{-1}$), and turbulent kinetic energy (TKE; m$^2$s$^{-2}$), expressed as a percentage of the range of observed values at TWR$_W$ (bottom row), for all D5 simulations. Statistics are computed from 40 1-min mean values, $\pm 20$ min from the time of peak 1-min mean observed heat flux: 15:38 EDT. For each variable, the three values correspond to the lower, middle, and upper tower levels, respectively. See Table 2 for description of simulations.

| Simulation | T | WS | VV | TKE |
|---|---|---|---|---|
| HFP$_{2.4L}$ | -12.3, -39.0, -43.3 | 14.8, -4.6, -7.6 | 27.4, 6.1, 12.4 | -18.3, -37.3, -38.5 |
| HFP$_{4.8L}$ | 0.6, -31.2, -40.9 | 30.3, 3.9, -2.5 | 27.5, 5.9, 11.6 | -12.1, -34.3, -37.0 |
| HFP$_{24L}$ | 65.7, 10.7, -26.1 | 89.1, 32.7, 7.2 | 26.3, 2.4, 6.1 | 23.5, -21.2, -32.6 |
| HFP$_{24H}$ | 12.9, -1.9, -13.3 | 84.7, 45.7, 17.5 | 24.2, -0.3, 4.2 | 17.0, -17.0, -29.0 |
| HP$_{400}$ | -13.2, -45.2, -44.0 | 3.7, -10.9, -8.9 | 27.3, 5.5, 11.3 | -19.9, -38.5, -38.0 |
| SHF$_{24}$ | 84.7, -7.6, -35.7 | 83.6, 30.8, 12.6 | 25.3, -0.3, 3.5 | 26.9, -20.0, -30.1 |
| NF$_0$ | -30.0, -49.3, -47.1 | -7.0, -14.2, -11.2 | 26.6, 5.3, 12.5 | -22.7, -37.9, -38.4 |
| O$_{max}$-O$_{min}$ | 16.8, 10.2, 7.6 | 1.4, 1.6, 3.3 | 0.9, 1.3, 1.8 | 3.3, 3.6, 8.4 |

all three statistics, all four variables and all three tower levels, HFP$_{24H}$ exhibits the smallest overall error (lowest percent MD and RMSD, and highest IA). Overall, the statistics indicate poorest agreement with observations for simulations with 100% of the peak 1-min mean observed heat flux concentrated at the surface or in the lowest few grid levels (SHF$_{24}$ and HFP$_{24L}$), in agreement with Sun et al. (2006). However even for HFP$_{24H}$, individual variables and levels exhibit percent MD and RMSD of 80-90% or higher and IA of about 0.25 (e.g., lower tower level wind speed).

# 6 Summary and Conclusions

In this study, we have examined different methods used to represent the sensible heat release from low-intensity prescribed fires in mesoscale models. Such fires are prevalent during prescribed fire operations in the eastern US and can impact the health and safety of both fire personnel and the general public. A series of simulations conducted with the ARPS and ARPS-CANOPY models have been evaluated using observations collected during a low-intensity prescribed fire in the NJPB. Although this study focused exclusively on low-intensity fires, the study findings may have relevance to mesoscale model simulations of higher-intensity fires, including wildfires.

A two-phase model assessment and sensitivity test approach was utilized in which a control simulation (HFP$_{2.4L}$) was examined first, followed by a comparison of simulations with different sensible heat source methods and parameters. The multi-dimensional model assessment confirmed that the model reproduced the background and fire-perturbed atmosphere as depicted by measurements made at the six towers. However, the model was found to underestimate the upper tower level TKE





**Table 4.** As in Table 3, but for root mean square difference.

| Simulation | T | WS | VV | TKE |
|---|---|---|---|---|
| HFP$_{2.4L}$ | 21.8, 45.6, 50.0 | 35.3, 34.9, 31.9 | 34.7, 26.5, 30.1 | 27.1, 45.4, 46.8 |
| HFP$_{4.8L}$ | 19.4, 39.5, 48.6 | 43.4, 33.7, 30.5 | 35.0, 26.9, 30.1 | 25.0, 44.3, 45.9 |
| HFP$_{24L}$ | 84.4, 47.4, 46.5 | 99.2, 48.1, 33.6 | 36.2, 31.1, 30.2 | 42.2, 38.2, 44.4 |
| HFP$_{24H}$ | 29.1, 32.2, 35.2 | 94.0, 56.0, 37.1 | 33.0, 25.9, 26.8 | 33.8, 33.8, 42.4 |
| HP$_{400}$ | 22.9, 50.7, 50.2 | 35.4, 36.9, 32.8 | 34.7, 26.0, 29.4 | 27.5, 46.0, 46.1 |
| SHF$_{24}$ | 106.1, 48.7, 50.5 | 94.3, 45.5, 35.6 | 35.6, 30.4, 27.5 | 43.9, 35.0, 42.7 |
| NF$_0$ | 36.5, 54.6, 52.5 | 34.6, 40.2, 33.6 | 33.7, 24.6, 29.1 | 29.8, 46.3, 46.8 |
| O$_{max}$-O$_{min}$ | 16.8, 10.2, 7.6 | 1.4, 1.6, 3.3 | 0.9, 1.3, 1.8 | 3.3, 3.6, 8.4 |

**Table 5.** As in Table 3, but for index of agreement (Willmott, 1981).

| Simulation | T | WS | VV | TKE |
|---|---|---|---|---|
| HFP$_{2.4L}$ | 0.54, 0.41, 0.35 | 0.38, 0.25, 0.39 | 0.41, 0.36, 0.38 | 0.40, 0.43, 0.45 |
| HFP$_{4.8L}$ | 0.70, 0.45, 0.35 | 0.37, 0.22, 0.36 | 0.41, 0.39, 0.37 | 0.36, 0.41, 0.45 |
| HFP$_{24L}$ | 0.32, 0.50, 0.35 | 0.25, 0.37, 0.32 | 0.36, 0.35, 0.38 | 0.31, 0.39, 0.45 |
| HFP$_{24H}$ | 0.61, 0.65, 0.52 | 0.27, 0.39, 0.33 | 0.40, 0.45, 0.47 | 0.41, 0.44, 0.45 |
| HP$_{400}$ | 0.55, 0.40, 0.35 | 0.39, 0.28, 0.39 | 0.40, 0.34, 0.36 | 0.42, 0.44, 0.46 |
| SHF$_{24}$ | 0.29, 0.41, 0.36 | 0.26, 0.39, 0.33 | 0.36, 0.38, 0.45 | 0.36, 0.45, 0.45 |
| NF$_0$ | 0.40, 0.37, 0.35 | 0.32, 0.27, 0.40 | 0.41, 0.37, 0.40 | 0.40, 0.42, 0.45 |

values at some of the towers. The model sensitivity tests revealed that the best agreement with observations occurred when the fire sensible heat release was represented as a turbulent sensible heat flux profile (method 1a) with 100% of the peak 1-min mean observed heat flux averaged across the in-situ flux towers and an e-folding extinction depth of 18 m (100% of the average canopy height in the burn unit) (HFP$_{24H}$). The poorest agreement was found when (i) 100% of the peak 1-min mean observed heat flux was concentrated at the surface or in the lowest few grid levels (SHF$_{24}$ and HFP$_{24L}$), or (ii) the peak 1-min mean observed heat flux value was heavily diluted (10-20% of peak 1-min mean observed value) before implementation in the model (HFP$_{2.4L}$ and HFP$_{4.8L}$).

The findings of this study have provided useful insight into the representation of the sensible heat source from a low-intensity fire in a mesoscale model. The findings suggest that methods that rely more heavily on the mesoscale model's native





land-surface and subgrid-scale turbulence parameterizations to move heat vertically through the near-surface atmosphere (SHF
and HFP with small e-folding extinction depth), can lead to a concentration of heat near the surface, excessively large near-
surface temperatures, and consequently an overestimation (underestimation) of near-surface (forest overstory) fire-perturbed
wind speeds and TKE. On the other hand, implementing a turbulent sensible heat flux in the model corresponding to 10-20%
of the peak 1-min mean observed value (i.e., heavily diluting the point heat flux measurement), as in Kiefer et al. (2014), can
lead to an overall underestimation of TKE near or just above the top of the forest overstory.

As with any study, limitations must be kept in mind when considering study findings and any potential applications. First,
the study findings derive from simulations performed with a single mesoscale model, ARPS-CANOPY, and it's parent model,
ARPS. It is expected that the findings will be applicable to other mesoscale models, but this will need be confirmed by the
broader model community, particularly for two-way coupled atmosphere-fire models like WRF-SFIRE. Although efforts to
couple a fire behavior model with ARPS-CANOPY are underway, the model at this time is still one-way coupled. Second, the
model simulations were evaluated with observations from a single low-intensity fire experiment. In future work, model simu-
lations will need to be evaluated using observations from fires spanning different intensities, in different forest environments,
and with different synoptic-mesoscale conditions. Third, this study examined only two extinction depths (4.4 and 18 m), with
a more complete examination of the extinction depth parameter space left to future work. Finally, the choice of e-folding ex-
tinction depth used in a particular mesoscale model simulation may ultimately depend on the intended model application. An
e-folding extinction depth that yields satisfactory agreement with flux tower measurements of mean TKE may prove too large
or too small in the context of smoke dispersion or fire behavior predictions (e.g., Kartsios et al., 2017; Kochanski et al., 2018).

*Code and data availability.* Data sets and software code associated with this study have been uploaded to the Harvard Dataverse (Kiefer
et al., 2021). Data are available under the terms of the Creative Commons Zero "No rights reserved" data waiver (CC0 1.0 Public domain
dedication).

*Author contributions.* The following authors participated in the conceptualization of this study: MTK, WEH, SZ, and JJC. The methodology
and numerical analyses were developed by MTK, and WEH, SZ, JJC, and XB participated in the interrogation and refinement of the analyses.
MTK was responsible for writing and preparing the original draft of the manuscript, and all co-authors provided review and editing. NSS,
KLC, MRG, JLH, and MP, were instrumental in the collection and preparation of the field data that provided the basis for validating the
results presented in this study. MTK served as the curator of the model study data archive.

*Competing interests.* The authors declare that they have no conflict of interest.





*Acknowledgements.* This research was funded by the US Department of Defense - Strategic Environmental Research and Development Program (SERDP) project #RC-2641. Special thanks are given to the New Jersey Forest Fire Service for conducting the burn, and Eric Mueller (National Institute of Standards and Technology) for processing the raw fire-tracker data. The color maps used in all figures except

Figs. 1, 2, and 5 were developed by Cynthia Brewer at the Pennsylvania State University (http://colorbrewer2.org/).





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
