# Peer review of "Representing Low-Intensity Fire Sensible Heat Output in a Mesoscale Atmospheric Model with a Canopy Submodel: A Case Study with ARPS-CANOPY (version 5.2.12)"

_Geoscientific Model Development, 2021_

## Author Comment (AC1)

Note to editor and reviewers:

1. Author answers are in red.  Our answer to each comment is broken down into (a) our response, and (b) a description of changes made to the text and/or figures.  *Page and line numbers correspond to the discussion preprint evaluated by the reviewers.*
2. The tracked changes PDF file was created using the Perl script *latexdiff* (https://ctan.org/pkg/latexdiff?lang=en).   The PDF file has been manually annotated to denote changes not highlighted by *latexdiff*.

**Reviewer #1**

Here is yet another paper, of a growing number of papers, which shows us breaking free the of the simple models of fire behaviour of the last 40 years.

This paper makes several contributions first and probably most important is that this is a controlled burn i.e., a management burn of low intensity and low rate of spread. Its mesoscale weather affect is rather limited compared to most of the published studies of large area fires with turbulent sensible heat fluxes that penetrate much higher into the atmosphere.

In general, I found the paper very detailed and complete in its discussion and interpretation of the model and data. It is an excellent example of how to organize and evaluate fire models with specific research question. I suspect we will hear more of the studies related to this fire. These approaches are the future so we must recruit managers and researchers who do not necessarily have any experience or limited experience with this now more mechanistic approach.

Response: Thank you for your review.  We agree that outreach and technology transfer are essential but feel that this is beyond the scope of this manuscript.
Changes:  N/A

It would have been useful for some more information on the distribution and density of the groundcover, shrubs and trees(height). Also on the amount of necrosis, mortality and litter and duff removed.

Response: We agree that some addition details are warranted regarding fuel loading and consumption.
Changes:  We have added a few sentences (at the end of line 122 on page 4) to describe fuel loading and consumption measured during the experiment: "Pre-burn loading of fine litter, woody fuels on the forest floor, and understory vegetation, estimated from 0.5 $m^2$ harvest plots, was $946 \pm 85$, $114 \pm 11$, and $287 \pm 28$ g $m^{-2}$ (mean $\pm$ one standard error; n = 12), respectively.  Post-burn loading of fine litter, woody fuels, and understory vegetation was $631 \pm 34$, $118 \pm 7$ and $261 \pm 36$ g $m^{-2}$, respectively, and consumption was estimated as 33%, -3%, and 9%, respectively; very little overstory fuel was consumed."

**Reviewer #2**

General comments:

The work presented in this manuscript implemented and compared three different fire sensible heat representations in a mesoscale weather model equipped with a canopy module. The work focused on low intensity prescribed fire scenarios which presented a timely study considering the need of such knowledge and tools for the fire risk management in the prescribed burn practices. The three fire sensible heat source

representations implemented in the ARPS-CANOPY mode will provide a great tool for future fire studies from mesoscale down to microscale.

Few efforts have been made to cross-compare and assess the suitability/limitation of these methods. Another important contribution of this work is the evaluation of these different fire representations. Utilizing extensive observational data from a field campaign, this work is able to evaluate different fire sensible heat source representation through sensitivity tests and the model-observation agreement.

In my opinion, this is a well written manuscript which presents not only the advancement in the numerical representation of the fire sensible heat source but also with sensitivity tests that provide guidelines for the future studies using the modelling tool. That being said, a few minor revisions might help the readers to better appreciate the work. See below for the minor issues.

Response: Thank you for your review.
Changes:  N/A

Line 50 to 56. I think the concept of the mesoscale model used here is slightly different from the traditional definition. Traditionally, most of the mesoscale model applications were done using pure parameterization with little or no resolved turbulence like Reynolds-averaged Navier–Stokes equations (RANS) rather than large eddy simulation (LES) approach. Recommend the author to add one or two sentences to clarify.

Response:  We agree that the mesoscale model application described in this study differs from traditional mesoscale model applications in the use of grid spacing that allows some scales of turbulent motion to be explicitly resolved.
Changes:  We added the following sentence after line 56 on page 2: "It is worth noting that the use of a mesoscale model with grid spacing that allows for some scales of turbulence to be explicitly resolved is a departure from traditional mesoscale model applications in which most or all turbulence is parameterized (Michioka and Chow, 2008; Kiefer et al., 2013; Chow et al., 2019)."

Table 1. The vertical extent/size of each domain should be provided.

Response:  Thank you for noting this oversight on our part.  We agree that this information is important to convey to the reader.
Changes:  In Table 1, we added the number of grid points in the vertical direction to the "Grid size" column, and the depth of the model domain to the "Domain size" column.  We added a sentence to the table caption to explain the order of dimensions in the "Grid size" and "Domain size" columns, and we added decimal points to values in the "Dx, Dy" column, to be consistent with the "Domain size" and "Dz min" columns.

Figure 5. The use of "x" in the figure to indicate minimum and maximum caused overlapping of the symbols in some of the subplots. Recommend to use a smaller "x". A

simple figure legend to highlight the observation versus modeled data might also be useful.

Response:  Thank you for your suggestion regarding the symbol sizes and the addition of a figure legend.
Changes: We reduced the size of all "X" symbols by 25% (both red and black symbols) and added a small legend to the upper-left panel; the figure caption was modified slightly to identify the legend.